# COVID-19 vaccine coverage among adults in Sarlahi District of Nepal in 2022

Porcia Manandhar[1]*, Joanne Katz[1], Tsering Pema Lama[2], Subarna K. Khatry[2], William J. Moss[1,3], Daniel J. Erchick[1]

1 Department of International Health, Johns Hopkins Bloomberg School of Public Health, Baltimore, Maryland, United States of America, 2 Nepal Nutrition Intervention Project-Sarlahi (NNIPS), Kathmandu, Nepal, 3 Department of Epidemiology, Johns Hopkins Bloomberg School of Public Health, Baltimore, Maryland, United States of America

* pmanand1@jhu.edu

## Abstract

Nepal launched its COVID-19 vaccination campaign in January 2021 through the COVID-19 Vaccines Global Access (COVAX) facility. Vaccine coverage, especially in low- and middle-income countries (LMICs), is measured using administrative-level data; however, this source is often subject to biases and limitations. We conducted a household survey in rural Sarlahi District, Nepal, to estimate COVID-19 vaccine coverage and assess associations with participant characteristics among adults. The quantitative household survey was conducted from August to December 2022 in four municipalities among 362 adults aged 18 years and older. The survey collected demographic data, vaccination status and vaccination accessibility details. Descriptive analyses included a summary of vaccination coverage, vaccine card availability, drop-out rate, and vaccine manufacturer. Multivariable regression modeling was used to analyze factors associated with completing the primary vaccination series. Three-quarters of participants (74.6%) completed at least the primary series (51.9% only completed the primary series, 22.7% were also boosted). Vaccine card retention was 86% among those with at least one dose. Odds of completing the primary series increased with age (31–50 yrs, adjusted odds ratio (aOR) = 3.07, 95% CI: (1.67, 5.8) and 51+years, aOR = 4.75, 95% CI: (2.06, 11.9) compared to 18–30 years). Wealthier groups had higher odds of completing the primary series than the poorest quartile (wealth quartile 2, aOR = 3.04, 95% CI: (1.41, 6.80); wealth quartile 3, aOR = 2.18, 95% CI: (1.05, 4.62); wealth quartile 4, aOR = 2.32, 95% CI: (1.06, 5.17)). Despite moderate primary series coverage and high card retention, booster coverage remained low. The population has exhibited a mix-and-match approach to COVID-19 vaccination, likely due to availability and accessibility. While the emergency stage of the pandemic has ended, lack of adequate vaccine coverage increases the immunity gap for a virus that continues to circulate and evolve.

**Data availability statement:** Deidentified data is attached with this submission.

**Funding:** Research reported in this publication was supported by the Sabin Vaccine Institute's Social and Behavioral grant (050165, awarded to DJE) and the Eunice Kennedy Shriver National Institute of Child Health and Human Development of the National Institutes of Health (Award number: R01HD109385, awarded to DJE). The funders had no role in study design, data collection and analysis, decision to publish, or preparation of the manuscript.

**Competing interests:** The authors have declared that no competing interests exist.

## Introduction

In January 2021, Nepal started its COVID-19 vaccination campaign with vaccine donations from India and China, and in March 2021 the country became one of the first to receive COVID-19 vaccines through the COVID-19 Vaccines Global Access (COVAX) facility [1]. COVAX, designed for equitable global access to COVID-19 vaccines, was a partnership between the Coalition for Epidemic Preparedness Innovations (CEPI), Gavi, the Vaccine Alliance, United Nations Children's Fund (UNICEF), and the World Health Organization (WHO) [1]. According to WHO, as of November 2023, 62.6 million doses of COVID-19 vaccines were administered to a population of ~30 million in Nepal [2]. By strictly following the Strategic Advisory Group of Experts on Immunization (SAGE)'s COVID-19 vaccination prioritization groups, the country started its vaccination campaign with frontline workers and the elderly population and then with age-group eligibility. COVID-19 booster doses were rolled-out starting in January 17, 2022 [3].

Childhood vaccines and tetanus toxoid (TT) vaccine during pregnancy have been well-accepted in Nepal, with coverage of 89% for the third dose of Diphtheria, tetanus, whooping cough (pertussis) or DTP, polio, hepatitis B and Haemophilus influenzae type b (Hib) (DTP-HepB-Hib), 89% for at least one dose of measles-rubella (MR) vaccines among children aged 12–23 months, and 93% for TT among women with a live birth, according to the Nepal Demographic Health Survey 2022 [4]. However, Nepal, like many low and middle-income countries (LMICs), does not have an adult vaccination platform beyond TT delivery through antenatal care services. In addition, Nepal struggled to keep steady access to COVID-19 vaccines due to the sporadic vaccine supply chain through the COVAX facility and global supply chain challenges, including syringe shortages and expiring stockpiles [5–7].

After the launch of Nepal's COVID-19 vaccination program, vaccine uptake was initially restricted by global availability, and therefore access was limited. Shortly after this time, Nepal instituted national and local vaccination mandates requiring proof of vaccination, with a physical vaccine card, to enter banks and receive social security for the elderly [8]. Vaccine demand and uptake also came to depend on perceived risks and vaccine hesitancy [9,10]. It is important to maintain coverage to prevent outbreaks given emerging SARS-CoV-2 variants and the continued risks for certain vulnerable populations, such as the elderly and pregnant women.

With one-third (34%) of the population living in rural areas of Nepal [11], measuring vaccine coverage in these settings is vital to assessing health system readiness, public trust in health authorities, and vaccine acceptance and hesitancy. A multi-country study in South Asia reported that urban communities had higher vaccine acceptance in India and Pakistan in contrast to Bangladesh where the rural population was more likely to accept a COVID-19 vaccine than urban participants [12]. Another study in rural Bangladesh showed that, despite low knowledge about COVID-19 vaccines, the population had a more favorable attitude towards vaccination [13]. In Maharashtra, India, tribal villagers, women, and those from lower socioeconomic status were less likely to have been vaccinated [14], while a different study in Tamil Nadu showed that

low educational attainment, low income and unemployment resulted in a higher likelihood of refusing the COVID-19 vaccine [15]. Rigorous studies have not estimated vaccine coverage in rural Nepal, especially the Terai region. We conducted a household survey to assess COVID-19 vaccine coverage and identify characteristics associated with coverage among adults in Sarlahi District, a mostly rural area in the southern plains of Nepal bordering Bihar, India.

## Methods

The study was conducted from August 8, 2022, through December 22, 2022.

### Study area

We conducted a quantitative household survey with adults aged 18 years and older. The households were identified from four Nagarpalikas (municipalities) and Gaunpalikas (rural municipalities) of an ongoing randomized trial, the Maternal Infant Nutrition Trial (MINT, National Clinical Trial registration number: NCT03668977), which was a community-based randomized trial of the impact of Balanced Energy Protein (BEP) supplements on various perinatal, infant, and maternal outcomes [16]. This study had previously constructed a population-based list of households in six Nagarpalikas and Gaunpalikas of rural Sarlahi District, Nepal, in which married females 15–30 years of age resided. We randomly selected households from this census, proportional to the ward-level population size, from 17 purposively selected wards in 4 Nagarpalikas and Gaunpalikas— Haripur, Ishworpur, Kabilasi and Chandranagar (Fig 1). We selected a sample size of 362 participants to allow for a two-sided 95% confidence interval with a width equal to 0.12 around an assumed sample prevalence of vaccine coverage of 50% [17].

### Recruitment and consenting

We approached the head of the household and explained their selection from the household census within the ongoing randomized trial. Then, using a household recruitment script, we asked for their permission to be recruited. If the head of household agreed to participate, he/she provided us with a list of adult household members (18 years and older) currently residing in the household. We then selected an adult household member at random using an offline random number generator on a data entry tablet. They were then asked if they wanted to participate in this one-time survey using a signed written consent form. If the selected household member was not present or did not consent, we selected another household member at random from the member list and continued to do so until we identified a household member who was present and agreed to participate.

Two study staff with experience in consenting and interviewing in Maithali and Nepali obtained oral consent from the head of household for the household to participate. Signed consent was obtained from the household member selected for the interview. The study staff conducted the interviews in person at the participant households using appropriate COVID-19 infection prevention protocols, including masking and social distancing.

### Measurements and definitions

The survey collected information on participant demographics, COVID-19 vaccination status, and vaccination details including vaccination card availability and vaccination dates and brands, along with modules on vaccine hesitancy, and vaccination accessibility.

Each participant's vaccination status was categorized into one of three groups—i) partially vaccinated (only one dose of a vaccine brand other than Janssen/Johnson & Johnson (J&J)), ii) fully vaccinated (one dose of J&J or a two-dose combination of other brands), iii) boosted (two doses of J&J or one dose of J&J and one dose of another brand or three doses of any combination of other brands) and unvaccinated [18]. If the vaccine brand was unknown and the first dose was missing, we classified the partial dose as missing, as determining partial vaccination status requires knowing the brand

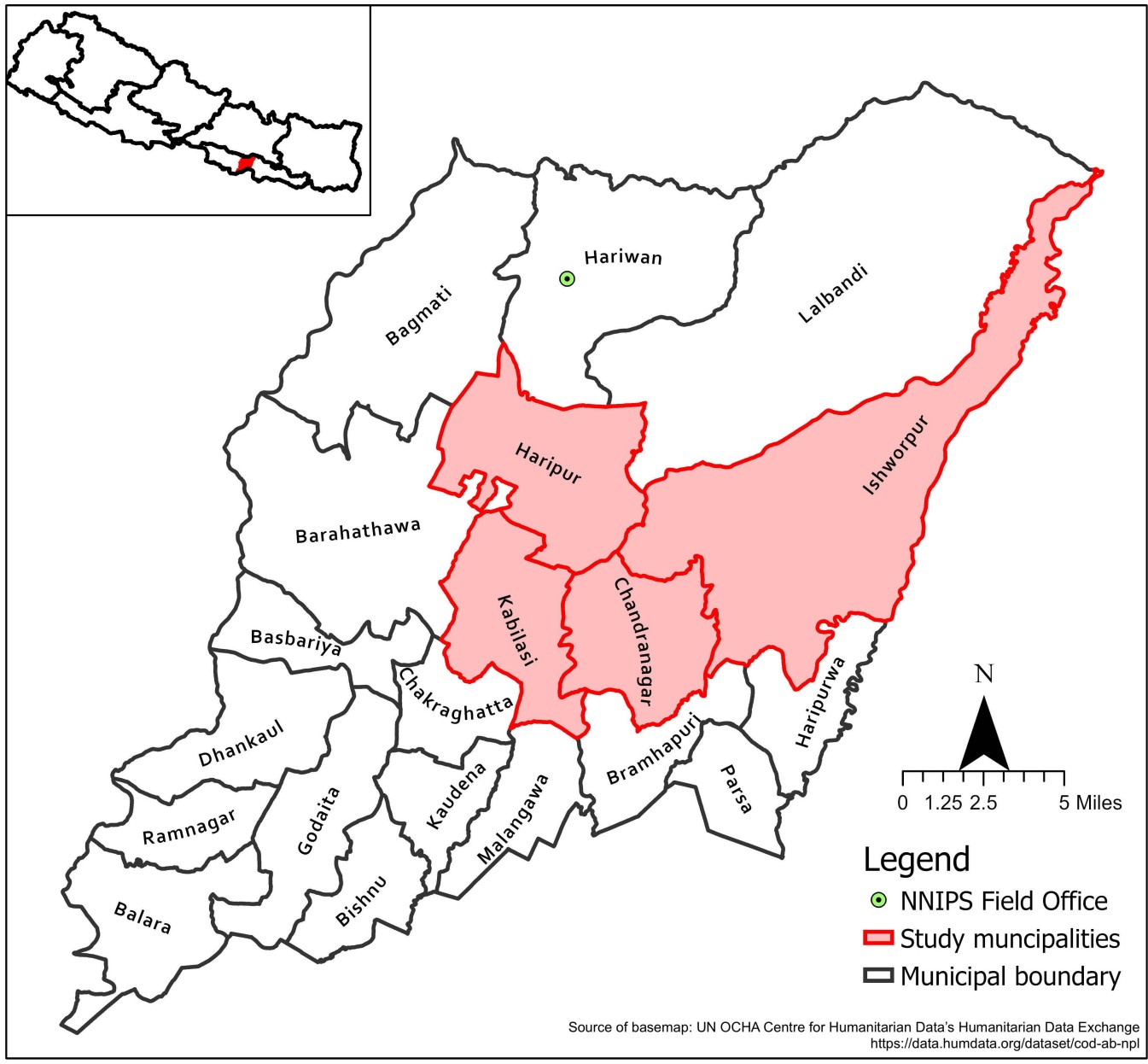

**Fig 1. Sarlahi district and the study municipalities (basemap link: https://data.humdata.org/dataset/cod-ab-npl).**

of at least the first dose. If the brand of either the first or second dose was missing, the full-dose status was considered missing, to avoid incorrect categorization. Finally, if all three doses were missing, the boosted status was also considered missing. Vaccine coverage for each of the groups was calculated by dividing the total number of people vaccinated within each group by the total study population interviewed.

We created a vaccination accessibility scale with possible values ranging from 0 to 5 based on the number of responses to five questions (S1 Table), including a participant's responses to facility distance being far, waiting time at the vaccine clinic being too long, being aware if the nearest health facility had vaccine stock, and losing income and having

personal expenses to obtain the vaccine (the vaccine itself was provided by the government for free) [17]. The values were then weighted by the number of questions answered to get a scale from 0 (least accessible) to 1 (most accessible). The final groupings were <0.50 (less accessible) or ≥0.50 (more accessible) for the area's vaccination facilities.

We created a wealth index based on twenty socio-economic questions relevant to this community, including those related to water source, presence of electricity, type of latrine use, land ownership, receipt of remittances from abroad, and ownership of household items and vehicles. These socio-economic questions have been used for prior studies in this population, including the parent study. We calculated an ordinal scale using responses from these questions ranging from 0 to 20 and divided scores into wealth quartiles.

### Data management and analysis

The survey was implemented through a REDCap application on password-protected tablets from which data were transferred to a secure cloud at the field site for storage. Data were analyzed using Stata Version 15 [19] and R Version 2023.12.0 + 369 [20]. All data utilized for this analysis are provided in S1 Data.

Descriptive analyses included summarizing vaccination coverage, vaccine card availability, drop-out rate, and vaccine brand combination frequency distributions. At the time of data collection, booster doses had been available for more than 6-months so booster dose receipt was included in the descriptive analysis [21,22]. Time-to-event analysis was also conducted using inverse Kaplan-Meier curves to compare the introduction and uptake of vaccine doses over time, and regression analysis. For the inverse Kaplan-Meier curves, the event was COVID-19 vaccination, which was stratified by number of doses, and observations were left censored.

A logistic regression model was run with the outcome being the odds of completing the primary series (fully vaccinated only and boosted against partially vaccinated and unvaccinated). For the regression covariates, age was grouped into three categories (≤30, 31–50, and 51 + years), education was grouped into four categories (no schooling, primary (grade 1–5), middle (grade 6–10), and higher secondary and tertiary (grade 11+)), and religion was grouped into Hindu and Islam. Christianity (n = 2) was treated as missing data and not included in the analysis. The regression models were assessed using the Hosmer-Lemoshow goodness-of-fit test.

### Ethics

The study received ethical approval from the Nepal Health Research Council (112/2022) and the Johns Hopkins Bloomberg School of Public Health Institutional Review Board (19794).

## Results

### Study population

We approached 365 adults family member from 362 households and interviewed 362 community members (one per household) after 3 refusals (Fig 2). Participants had a median age of 35 years (IQR = 23.8) ranging from 18 to 86 years (Table 1). The majority were female (78.2%) and about half had no formal education (53.3%) (Table 1). Most were from the Madeshi (92.8%) ethnic group and were Hindu (81.8%) or Muslim (17.7%). Geographically, 18.5% of the study population resided in the Nagarpalika of Chandranagar, 35.1% in Haripur, 29.8% in Ishworpur, and 16.6% in Kabilasi. Almost all participants had mobile phone accessibility (96.1%) and electricity (97.0%) in their households. Thirty-two percent had a family member working abroad. Population characteristics by vaccination status are provided in Table 1.

### Vaccination status and vaccine card availability

Of the total participants, 74.6% had completed the primary series (51.9% completed the primary series only and another 22.7% were also boosted) (Table 2). Among those who completed the primary series, 30.4% (n = 82 of 270) did not receive

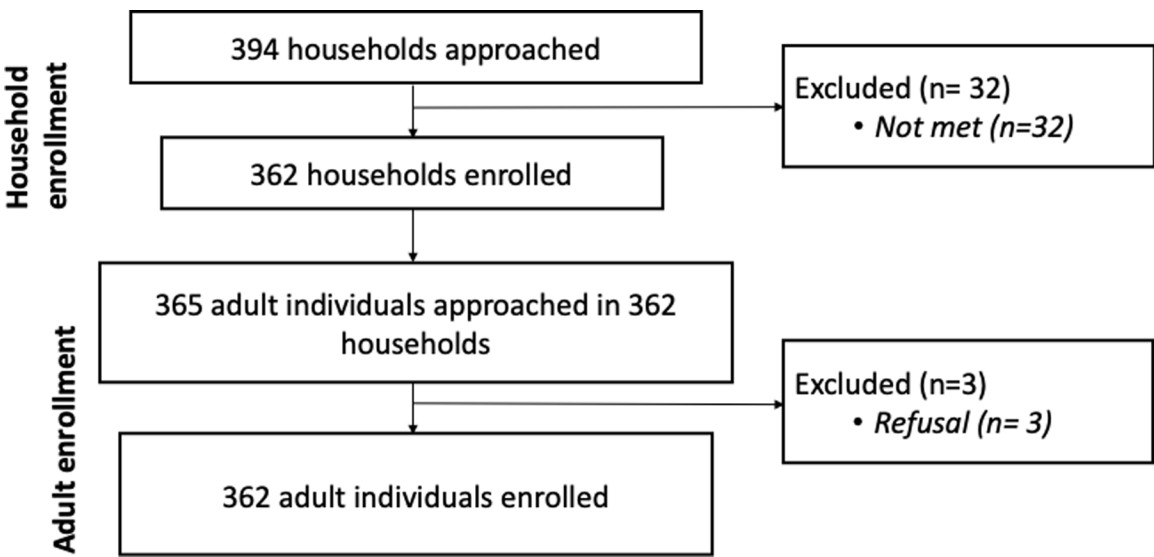

**Fig 2. Flowchart of study population recruitment.**

a booster dose. Of the 11.3% who were unvaccinated, 90.2% reported that they would be open to receiving a COVID-19 vaccine in the future [17]. Among the vaccinated, 86% had their vaccination cards available (Table 2).

The median age of vaccination for the participants who completed the primary series was 37.0 years (range = 18–82 years), which was older than those who did not complete the primary series (25.0 (range = 18–86) years) (p value <0.001, Table 1).

## Vaccine brand prevalence and preference

Most of the participants received Verocell (Sinopharm BBIBP manufactured in China) (38.6%) or Covishield (30.9%) (ChAdOx1 nCoV-19, licensed from Oxford AstraZeneca (AZD222) and manufactured in India), as their first dose of vaccine amongst the vaccinated (n = 321) followed by J&J (13.2%) (Table 3). Of the 236 who received a second dose of vaccine, the majority received Verocell (52.1%) or Covishield (41.1%) (Table 3). Three participants also received CoronaVac and Sputnik v for their first and second doses, which had not been distributed in the country and were obtained abroad (Table 3). The second vaccine dose was different than the first for 8.6% participants (n = 209; excluding 9 participants who received two doses either did not know their first or their second dose or both). The brands and dose combinations received are presented in Fig 3 and S2 Table.

Of all vaccinated individuals, only 12.7% (n = 46) stated that they had a preference for a particular vaccine brand for their first dose even if they did not end up receiving their preference. Of the individuals who preferred a specific brand for their first dose, 89% (n = 41) preferred J&J.

## Factors associated with vaccination status

The odds of completing the primary series increased with age (31–50 yrs, adjusted odds ratio (aOR) = 3.07, 95% confidence interval (CI): (1.67, 5.8) and 51+ yrs, aOR = 4.75, 95% CI: (2.06,11.9) in comparison to the youngest group 18–30 years old) and wealth group (wealth quartile 2, aOR = 3.04, 95% CI: (1.41,6.80); wealth quartile 3, aOR = 2.18, 95% CI: (1.05, 4.62); wealth quartile 4, aOR = 2.32, 95% CI: (1.06, 5.17) in comparison to the poorest wealth quartile, wealth quartile 1) (Table 4).

**Table 1.** Characteristics of the study population by completion of primary series.

| Characteristics | Overall[2] (N = 362) | Incomplete Primary series[2] (Under vaccinated) (N = 92) | Complete Primary series[2] (Fully vaccinated) (N = 270) | p-value[3] |
|---|---|---|---|---|
| **Age[1]** | 35.0 [23.8] | 25.0 [19.3] | 37.0 [22.5] | <0.001 |
| **Age groups** | | | | <0.001 |
| 18-20 | 42 (11.6%) | 29 (32%) | 13 (4.8%) | |
| 21-30 | 112 (30.9%) | 32 (35%) | 80 (30%) | |
| 31-40 | 78 (21.5%) | 11 (12%) | 67 (25%) | |
| 41-50 | 53 (14.6%) | 11 (12%) | 42 (16%) | |
| 51-60 | 37 (10.2%) | 5 (5.4%) | 32 (12%) | |
| 61+ | 40 (11.0%) | 4 (4.3%) | 36 (13%) | |
| **Sex** | | | | 0.200 |
| Female | 283 (78.2%) | 76 (83%) | 207 (77%) | |
| Male | 79 (21.8%) | 16 (17%) | 63 (23%) | |
| **Education level** | | | | 0.300 |
| No formal education | 193 (53.3%) | 44 (48%) | 149 (55%) | |
| Primary (grade 1–5) | 54 (14.9%) | 15 (16%) | 39 (14%) | |
| Middle (grade 6–8) | 37 (10.2%) | 14 (15%) | 23 (8.5%) | |
| Secondary (grade 9–10) | 35 (9.7%) | 7 (7.6%) | 28 (10%) | |
| Higher-Secondary (grade 11–12) | 37 (10.2%) | 11 (12%) | 26 (9.6%) | |
| Tertiary (grade 12+)* | 6 (1.7%) | 1 (1.1%)* | 5 (1.9%)* | |
| **Ethnicity** | | | | 0.092 |
| Madeshi | 336 (92.8%) | 89 (97%) | 247 (91%) | |
| Pahadi | 26 (7.2%) | 3 (3.3%) | 23 (8.5%) | |
| **Religion** | | | | 0.074 |
| Christian* | 2 (0.6%)* | 0 (0%)* | 2 (0.7%)* | |
| Hindu | 296 (81.8%) | 70 (76%) | 226 (84%) | |
| Islam | 64 (17.7%) | 22 (24%) | 42 (16%) | |
| **Water source** | | | | >0.900 |
| Tube well or tap | 361 (99.7%) | 92 (100%) | 269 (100%) | |
| Unprotected ring well* | 1 (0.3%)* | 0 (0%)* | 1 (0.4%)* | |
| **Latrine type** | | | | 0.039 |
| Brick and cement latrine | 320 (88.4%) | 75 (82%) | 245 (91%) | |
| No latrine present | 33 (9.1%) | 14 (15%) | 19 (7.0%) | |
| Pit latrine | 9 (2.5%) | 3 (3.3%) | 6 (2.2%) | |
| **Mobile phone access** | 348 (96.1%) | 87 (95%) | 261 (97%) | 0.400 |
| **Electricity** | 351 (97.0%) | 89 (97%) | 262 (97%) | >0.9 |
| **Vaccination accessibility** | | | | 0.300 |
| Vaccination less accessible | 42 (11.6%) | 8 (8.7%) | 34 (13%) | |
| Vaccination more accessible | 320 (88.4%) | 84 (91%) | 236 (87%) | |
| **Family working abroad** | | | | 0.900 |
| Do not know* | 1 (0.3%)* | 0 (0%)* | 1 (0.4%)* | |
| No | 244 (67.4%) | 63 (68%) | 181 (67%) | |
| Yes | 117 (32.3%) | 29 (32%) | 88 (33%) | |
| **Social media use** | 196 (54.1%) | 46 (50.0%) | 150 (55.6%) | 0.400 |

*(Continued)*

**Table 1.** (Continued)

| Characteristics | Overall[2] (N=362) | Incomplete Primary series[2] (Under vaccinated) (N=92) | Complete Primary series[2] (Fully vaccinated) (N=270) | p-value[3] |
|---|---|---|---|---|
| **Wealth quartiles** | | | | 0.005 |
| 1 (Poorest) | 91 (25.1%) | 35 (38%) | 56 (21%) | |
| 2 | 91 (25.1%) | 15 (16%) | 76 (28%) | |
| 3 | 90 (24.9%) | 23 (25%) | 67 (25%) | |
| 4 (Wealthiest) | 90 (24.9%) | 19 (21%) | 71 (26%) | |

[1]Median (IQR);

[2]n (%);

[3]Pearson's Chi-squared test or Fisher's exact test.

*Excluded from statistical independence test.

No missing data present.

**Table 2. Vaccination card availability by level of vaccine coverage among vaccinated individuals.**

| Vaccination groups | Card availability (n, %) | | |
|---|---|---|---|
| | No (N=45) | Yes (N=276) | Overall (N=321) |
| One-dose only | 12 (26.7%) | 39 (14.1%) | 51 (15.9%) |
| Primary series only | 30 (66.7%) | 158 (57.2%) | 188 (58.6%) |
| Primary series and boosted | 3 (6.7%) | 79 (28.6%) | 82 (25.5%) |

## Timeliness of vaccine doses

The uptake of the first and second doses rose quickly in the first year but plateaued around 85.6% and 62.7% respectively (Fig 4). The booster dose was introduced in January 2022 and its uptake was not as steep as the other doses and soon stagnated (Fig 4) during the time of the survey.

## Discussion

Approximately three-fourths of participants competed the primary COVID-19 vaccination series towards the end of 2022 in our study. In terms of booster dose, Nepal had a coverage ranging from 21.3% to 28.4% from July 1 to December 1 2022 [23], similar to our estimate in the four Sarlahi municipalities. A study conducted in the urban slums of India from December 2022 to March 2023 reported a booster coverage of 58.3%, much higher than our rural study population's coverage.

In the early days of the pandemic, vaccination mandates, in addition to national recommendations, could be a potential reason for the higher completion of the primary series in comparison to the booster doses in Sarlahi District. By the time booster doses were rolled out in 2022, mandates were not being strongly enforced by the government or local institutions. In a study in India, the likelihood of receiving a booster doses increased with older age, higher education, and higher income [24]. Another study, conducted through a web-based survey in all 28 states of India, reported that healthcare professionals' trust in their government recommendations increased their confidence in accepting a COVID-19 booster dose [25]. In a multi-country study in Bangladesh, India, and Nepal, participants were more likely to be vaccinated if it was recommended by the government [12]. In our study population, we found high trust in the government's vaccine roll out plan and in local healthcare workers [17]. In high-income countries like the US and Canada, booster dose uptake was lower by

**Table 3. Prevalence of COVID-19 vaccine brands for different vaccine doses.**

| Dose brand | n (%) | Overall (N) |
|---|---|---|
| **Dose 1 brand** | | |
| Verocell vaccine(Sinopharm BBIBP/ China) | 140 (38.6%) | 321 |
| Covishield (AstraZeneca/ India) vaccine | 112 (30.9%) | |
| Johnson & Johnson | 48 (13.2%) | |
| Do not know | 10 (2.8%) | |
| Moderna | 7 (1.9%) | |
| CoronaVac | 2 (0.6%) | |
| Sputnik v | 1 (0.3%) | |
| Pfizer | 1 (0.3%) | |
| **Dose 2 brand** | | |
| Verocell vaccine (Sinopharm BBIBP / China) | 123 (52.1%) | 236 |
| Covishield (AstraZeneca/ India) vaccine | 97 (41.1%) | |
| Do not know | 7 (3.0%) | |
| Moderna | 3 (1.3%) | |
| Johnson & Johnson | 2 (0.85%) | |
| CoronaVac | 2 (0.85%) | |
| Sputnik v | 1 (0.4%) | |
| Pfizer | 1 (0.4%) | |
| **Dose 3 brand** | | |
| Covishield (AstraZeneca/ India) vaccine | 41 (59.4%) | 69 |
| Moderna | 14 (20.3%) | |
| Verocell vaccine(Sinopharm BBIBP/ China) | 10 (14.5%) | |
| Pfizer | 3 (4.3%) | |
| Do not know | 1 (1.4%) | |

No missing data present.

race/ethnic disparities, lower income, and education, lack of insurance coverage, immigration status, and having previous COVID-19 infection [26,27].

Vaccination card retention for COVID-19 vaccines for adults in our study was high in comparison to routine childhood immunization cards in Nepal (79% for children age 12–23 months and 61% for children aged 24–35 months) [28]. During field visits, we noticed many individuals kept their vaccination cards safely with land ownership, citizenship, and other important documents. Additionally, migrant labor destinations like the Middle East, a common destination for Nepali workers to seek employment (especially from Madhesh province within which Sarlahi is located [29]), imposed mandatory vaccination, preferably the one-shot J&J for immediate travel [30]. This encouraged workers to receive COVID-19 vaccines and retain vaccination cards for vaccination certification [31–33]. This is likely why vaccination card retention was much higher than seen with childhood immunizations. A study among migrant workers in Saudi Arabia reported South Asian workers being more likely to accept (93.1%) the vaccine upon its approval compared with Middle Easterners [34]. Additionally, despite vaccine accessibility being reported as high in our study population, the majority did not know if vaccines were available in their nearest vaccination facility, indicating poor communication [17].

The study population accepted heterologous (or "mix and match") vaccines from different vaccination brands and technologies for their second and booster shots. While the vaccination brand decision may have driven mostly by the

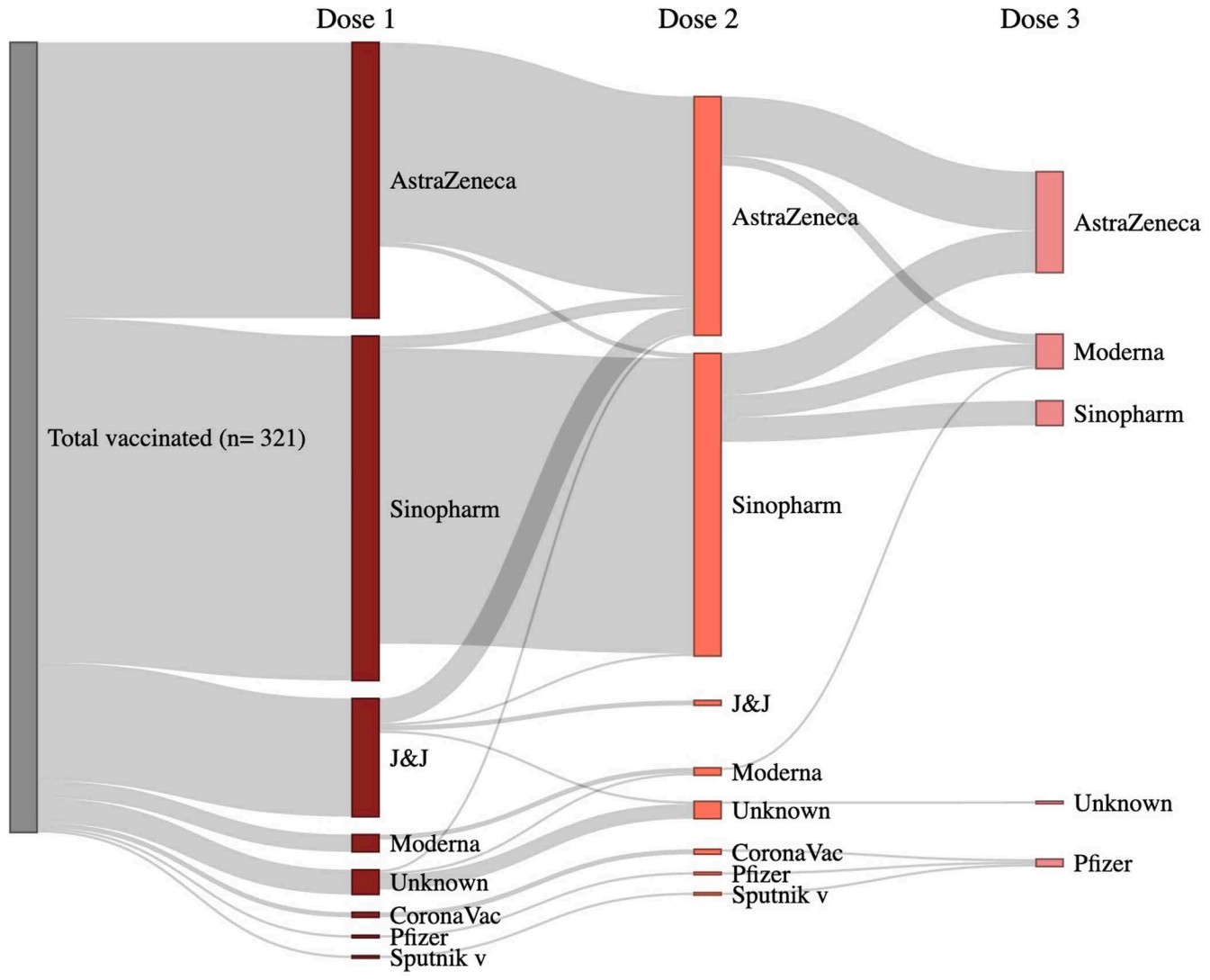

**Fig 3. COVID-19 vaccine brand combination amongst vaccinated individuals.**

availability of stock based on political donations and the global supply from the COVAX program in Nepal, mixing brands/technologies is safe and effective and potentially more immunogenic [35]. Heterologous vaccination, especially in poorer regions, was necessary due to global shortages and infections due to emerging new variants. However, it is a safe and efficient strategy for a sustainable vaccination program. Bhutan, due to the temporary export halt of the AstraZeneca COVID-19 vaccine from India, decided to use mRNA vaccines, Moderna and Pfizer [36], for their second dose to complete the primary series. They were able to vaccinate more than 95% of their eligible population in the first two rounds of a vaccination campaign in 2021 [37].

One limitation of our study was the cross-sectional design that only captured the state of vaccination during a particular stage of the pandemic. The study took place after the spillover of the Delta variant infections from neighboring India and

**Table 4. Logistic regression results for completing primary series (fully vaccinated).**

| Participant characteristic | n (%) | | Unadjusted | | | Adjusted | | |
|---|---|---|---|---|---|---|---|---|
| | Complete Primary series (Fully vaccinated) (N = 270) | Incomplete Primary series (Under vaccinated) (N = 92) | Odds ratio[1] | 95% CI[1] | | Odds ratio[1] | 95% CI[1] | |
| **Age ≤ 30 yrs** | 93 (34.4%) | 61 (66.3%) | **Ref** | | | | | |
| 31-50 yrs | 109 (40.4%) | 22 (23.9%) | **3.25** | **1.88** | **5.79** | **3.07** | **1.67** | **5.80** |
| 51 + years | 68 (25.2%) | 9 (9.8%) | **4.96** | **2.40** | **5.79** | **4.75** | **2.06** | **11.9** |
| **Female** | 207 (76.7%) | 76 (82.6%) | **Ref** | | | | | |
| Male | 63 (23.3%) | 16 (17.4%) | 1.45 | 0.80 | 2.73 | 1.15 | 0.57 | 2.37 |
| **No schooling** | 149 (55.2%) | 44 (47.8%) | **Ref** | | | | | |
| Primary (grade 1–5) | 39 (14.4%) | 15 (16.3%) | 0.77 | 0.39 | 1.55 | 0.70 | 0.32 | 1.57 |
| Middle (grade 6–10) | 51 (18.9%) | 21 (22.8%) | 0.72 | 0.39 | 1.33 | 0.67 | 0.31 | 1.45 |
| Higher-Secondary and tertiary (grade 11+) | 31 (11.5%) | 12 (13.0%) | 0.76 | 0.37 | 1.66 | 0.88 | 0.35 | 2.31 |
| **Hindu*** | 226 (83.7%) | 70 (76.1%) | **Ref** | | | | | |
| Islam | 42 (15.6%) | 22 (23.9%) | 0.59 | 0.33 | 1.07 | 0.53 | 0.27 | 1.04 |
| **Vaccination less accessible (<0.50 score)** | 34 (12.6%) | 8 (8.7%) | **Ref** | | | | | |
| Vaccination more accessible (≥0.5 score) | 236 (87.4%) | 84 (91.3%) | 0.66 | 0.28 | 1.42 | 0.69 | 0.27 | 1.59 |
| **No social media** | 120 (44.4%) | 46 (50.0%) | **Ref** | | | | | |
| Uses social media | 150 (55.6%) | 46 (50.0%) | 1.25 | 0.78 | 2.01 | 1.29 | 0.71 | 2.35 |
| **Wealth quartile 1 (poorest)** | 56 (20.7%) | 35 (38.0%) | **Ref** | | | | | |
| Wealth quartile 2 | 76 (28.1%) | 15 (16.3%) | **3.17** | **1.60** | **6.50** | **3.04** | **1.41** | **6.80** |
| Wealth quartile 3 | 67 (24.8%) | 23 (25.0%) | 1.82 | 0.97 | 3.47 | **2.18** | **1.05** | **4.62** |
| Wealth quartile 4 (richest) | 71 (26.3%) | 19 (20.7%) | **2.34** | **1.22** | **4.58** | **2.32** | **1.06** | **5.17** |

[1]OR = Odds Ratio, CI = Confidence Interval.

*Christianity (n = 2) was excluded from the regression analysis.

Bolded values are significant (p < 0.05).

escalating mortality [38,39]. The study was conducted in four municipalities, which roughly approximate the larger Terai region, but is not necessarily representative of Nepal overall. This study utilized a pre-existing census from the parent randomized trial as the sampling frame that comprised households that had a woman 15–30 years of age. Hence, our study households may not reflect all households in the district, although most households, which include multigenerational families, have women of childbearing age. Our study population had a higher proportion of females, as males tended to be working (farming or other) during the day or working overseas. Hence, this gender imbalance may contribute to a lack of generalizability. Moving forwards, population surveys like the MICS and DHS could add a COVID-19 vaccination module to capture this information.

The study had several strengths. There was a sufficient sample size to allow for a reliable estimate of coverage in this population with a high response rate. The study population composition included older individuals, and lower literacy, who are not always represented in online surveys [40,41]. The study population also represented agrarian households with families living and working abroad. The household surveys allowed us to confirm vaccination status by reviewing the vaccination cards, so coverage estimates are likely to be accurate and not influenced by social desirability bias.

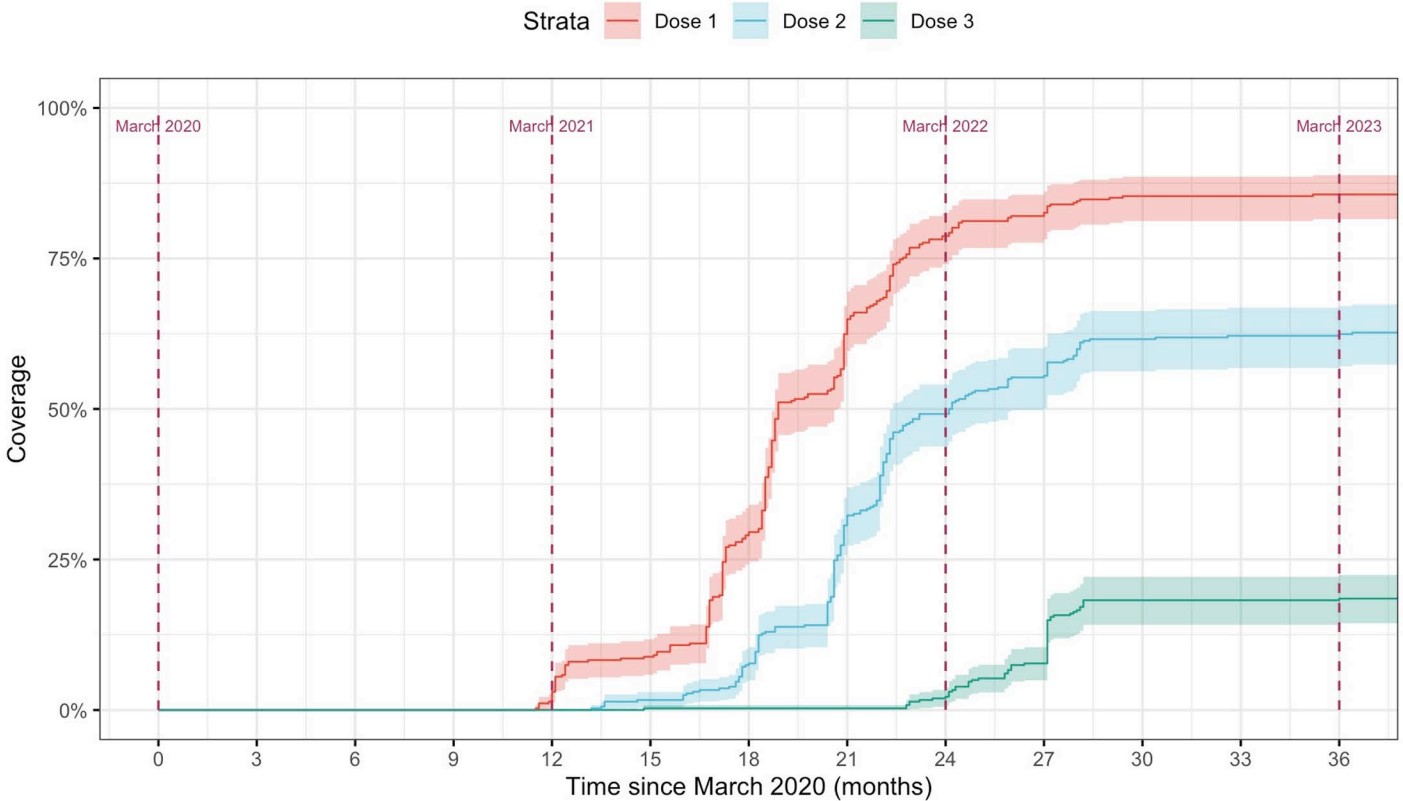

**Fig 4. Timeliness comaprision of COVID-19 vaccine doses.**

## Conclusion

In this rural sample of adults, there was moderate coverage for primary-series COVID-19 vaccination along with high vaccine card retention. However, booster dose coverage was low at the time of the study as booster doses were only recently introduced in the area. The population mixed vaccine brands, likely due to availability and accessibility. Interventions that encourage a primary series and booster dose uptake are needed at the national, provincial, and local to improve accessibility, demand, and vaccine coverage in rural areas.

## Supporting information

**S1 Table. Vaccination accessibility questions among community members.**
(DOCX)

**S2 Table. COVID-19 vaccine brand combinations.**
(DOCX)

**S1 Data. Data collected and used for analysis.**
(CSV)

## Author contributions

**Conceptualization:** Porcia Manandhar, Joanne Katz, Tsering Pema Lama, Daniel J. Erchick.

**Data curation:** Porcia Manandhar.

**Formal analysis:** Porcia Manandhar.

**Funding acquisition:** Porcia Manandhar, Joanne Katz, Daniel J. Erchick.

**Investigation:** Porcia Manandhar.

**Methodology:** Porcia Manandhar, Joanne Katz.

**Project administration:** Porcia Manandhar, Tsering Pema Lama, Subarna K Khatry, Daniel J. Erchick.

**Resources:** Porcia Manandhar.

**Supervision:** Joanne Katz, Tsering Pema Lama, Daniel J. Erchick.

**Visualization:** Porcia Manandhar.

**Writing – original draft:** Porcia Manandhar, Joanne Katz.

**Writing – review & editing:** Porcia Manandhar, Joanne Katz, Tsering Pema Lama, William J. Moss, Daniel J. Erchick.

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
