## [Decision Letter · Decision Letter 0]

22 Oct 2024

PGPH-D-24-02053

COVID-19 vaccine coverage in Sarlahi District of Nepal

Dear Dr. Manandhar,

Thank you for submitting your manuscript to PLOS Global Public Health. After careful consideration, we feel that it has merit but does not fully meet PLOS Global Public Health’s publication criteria as it currently stands. Therefore, we invite you to submit a revised version of the manuscript that addresses the points raised during the review process.

We look forward to receiving your revised manuscript.

Kind regards,

Brian Wahl

Academic Editor

Journal Requirements:

1. Please include a complete copy of PLOS’ questionnaire on inclusivity in global research in your revised manuscript. Our policy for research in this area aims to improve transparency in the reporting of research performed outside of researchers’ own country or community. The policy applies to researchers who have travelled to a different country to conduct research, research with Indigenous populations or their lands, and research on cultural artefacts. The questionnaire can also be requested at the journal’s discretion for any other submissions, even if these conditions are not met. Please find more information on the policy and a link to download a blank copy of the questionnaire here: https://journals.plos.org/globalpublichealth/s/best-practices-in-research-reporting. Please upload a completed version of your questionnaire as Supporting Information when you resubmit your manuscript

2. In the online submission form, you indicated that [Upon request]. 

a. In a public repository, 

b. Within the manuscript itself, or 

c. Uploaded as supplementary information.

3. Please provide separate figure files in .tif or .eps format.

4. Figure 3: please (a) provide a direct link to the base layer of the map (i.e., the country or region border shape) and ensure this is also included in the figure legend; and (b) provide a link to the terms of use / license information for the base layer image or shapefile. We cannot publish proprietary or copyrighted maps (e.g. Google Maps, Mapquest) and the terms of use for your map base layer must be compatible with our CC-BY 4.0 license. 

Additional Editor Comments (if provided):

Reviewers' comments:

Reviewer's Responses to Questions

**Comments to the Author**

1. Does this manuscript meet PLOS Global Public Health’s publication criteria ? Is the manuscript technically sound, and do the data support the conclusions? The manuscript must describe methodologically and ethically rigorous research with conclusions that are appropriately drawn based on the data presented.

Reviewer #1: Yes

Reviewer #2: Partly

2. Has the statistical analysis been performed appropriately and rigorously?

Reviewer #1: Yes

Reviewer #2: No

3. Have the authors made all data underlying the findings in their manuscript fully available (please refer to the Data Availability Statement at the start of the manuscript PDF file)?

Reviewer #1: Yes

Reviewer #2: Yes

4. Is the manuscript presented in an intelligible fashion and written in standard English?

Reviewer #1: Yes

Reviewer #2: No

5. Review Comments to the Author

Reviewer #1: The study which is descriptive is simple and easy to understand. The study evaluates COVID-19 vaccination coverage and determinants that could be relevant in similar vaccine preventable epidemics. The authors also outlined the limitations of the study. I think it may have added value to the study if the occurrence of COVID-19 infection among the study participants was investigated to determine vaccination effectiveness.

Reviewer #2: Comments

Title

1. Population of interest (rural adults/children), timeline component and type of study should preferably be mentioned in the title.

Abstract

1. Methods – Repeating of variables is there (for example: Vaccine card availability is repeated twice); Instead something like “Data was summarized using mean (SD) and frequency with proportions” could be mentioned. And the word “risk factors” is not appropriate here. Better to frame it as “factors associated with”

Introduction

1. Justification for the methodology adopted in the study needs to improve - why assessing COVID – 19 vaccine coverage in the first place? Why restricted only to rural area? Why coverage in children not assessed?

2. References needed for the following paragraph – “As vaccine access increased, demand and uptake came to depend not just on access but also on perceived risks and vaccine hesitancy. It is important to maintain coverage to prevent outbreaks given emerging SARS-CoV-2 variants and the continued risks for certain vulnerable populations. such as the elderly and pregnant women.”

Methods

1. Time period – Can mention the timeframe only. The statement “after the spillover of the Delta variant infections from neighboring India and escalating mortality (14,15)” can be shifted to discussion.

2. Sampling –

• How did the authors sample these four nagarpalikas/gaunpalikas from the entire Sarlahi district? Is it representative of the entire district?

• How the households were selected from each ward?

• Better to show the multistage cluster sampling which has been adopted using a flowchart with sampling technique adopted at every stage from the district to the individual level.

3. Sample size –

• What is the estimated prevalence for vaccine coverage (not vaccine hesitancy) taken for the study? Prevalence should be 50% (not 0.5).

• What is the absolute/relative precision used?

• Kindly cross check the calculation and rewrite it appropriately.

4. Recruitment and consenting –

• Could the timing of data collection have influenced which gender is more likely to be recruited? (females are more here)

• Were the questions self administered or interviewer administered?

• Whether efforts were taken to include those that were “Not met (n=32)”

• The writing in this section has scope of improvement with respect to grammar and by avoiding repetitions of sentences.

5. Measurement and definitions –

• Why was it decided to categorize vaccination status as partially if other brands were administered? Kindly provide justification for the classification used. If the participants didn’t know the brand, how were they classified?

• Some of the parameters used to calculate vaccination accessibility don’t belong to this category (For example: “being aware if the nearest health facility had vaccine stock” comes under knowledge more than accessibility). Was this scale derived from a standard tool? If yes, provide details for the same.

6. Data management and analysis –

• Kindly mention how descriptive analysis was done – mean (SD) or frequency with proportions

• For inferential statistics – 95% CI and p value (how much is significant)

• Regression – Better to report prevalence odds rather than odds ratio as this is a cross sectional study

Results

1. Table 1: The variables “Water source/latrine type”, etc are already accounted for in wealth quartiles. Mentioning these variables separately might not be appropriate while seeing for difference between completed vs incomplete primary series. If the author wants to retain these variables, they can create a separate table for basic socio-demographic characteristics and mention it there without comparing completed vs incomplete primary series. They can then include a few selected variables for the comparison aspect.

2. Justification for including table 2 (Is there a need to highlight a separate table for vaccine card availability?)

3. Table 4: – Better to report prevalence odds; How was social media usage measured? Has this variable been mentioned in earlier tables?

Discussion

1. There is scope of improvement in the structuring and flow of content in discussion.

2. Details related to vaccine accessibility could be discussed with respect to other studies

3. Content related to country of Bhutan can be cut short and relevant points to your study could be retained

4. Other biases with respect to sampling and measurement could be elaborated and its implication discussed.

6. PLOS authors have the option to publish the peer review history of their article (what does this mean? ). If published, this will include your full peer review and any attached files.

**Do you want your identity to be public for this peer review?** For information about this choice, including consent withdrawal, please see our Privacy Policy .

Reviewer #1: No

Reviewer #2: No

---

## [Decision Letter · Decision Letter 1]

28 Mar 2025

COVID-19 Vaccine Coverage Among Adults in Sarlahi District of Nepal in 2022

PGPH-D-24-02053R1

Dear Dr. Manandhar,

We are pleased to inform you that your manuscript 'COVID-19 Vaccine Coverage Among Adults in Sarlahi District of Nepal in 2022' has been provisionally accepted for publication in PLOS Global Public Health.

Best regards,

Brian Wahl

Academic Editor

Reviewer Comments (if any, and for reference):

Reviewer's Responses to Questions

**Comments to the Author**

1. If the authors have adequately addressed your comments raised in a previous round of review and you feel that this manuscript is now acceptable for publication, you may indicate that here to bypass the “Comments to the Author” section, enter your conflict of interest statement in the “Confidential to Editor” section, and submit your "Accept" recommendation.

Reviewer #2: All comments have been addressed

2. Does this manuscript meet PLOS Global Public Health’s publication criteria ? Is the manuscript technically sound, and do the data support the conclusions? The manuscript must describe methodologically and ethically rigorous research with conclusions that are appropriately drawn based on the data presented.

Reviewer #2: Partly

3. Has the statistical analysis been performed appropriately and rigorously?

Reviewer #2: Yes

4. Have the authors made all data underlying the findings in their manuscript fully available (please refer to the Data Availability Statement at the start of the manuscript PDF file)?

Reviewer #2: Yes

5. Is the manuscript presented in an intelligible fashion and written in standard English?

Reviewer #2: Yes

6. Review Comments to the Author

Reviewer #2: The authors have addressed majority of the comments.

7. PLOS authors have the option to publish the peer review history of their article (what does this mean? ). If published, this will include your full peer review and any attached files.

**Do you want your identity to be public for this peer review?** For information about this choice, including consent withdrawal, please see our Privacy Policy .

Reviewer #2: No
